# Project Management Maturity and Business Excellence in the Context of Industry 4.0

Angela Fajsi *, Slobodan Morača, Marko Milosavljević and Nenad Medić

Faculty of Technical Sciences, University of Novi Sad, 21000 Novi Sad, Serbia; moraca@uns.ac.rs (S.M.); milosavljevic.di12.2018@uns.ac.rs (M.M.); medic.nenad@uns.ac.rs (N.M.)

* Correspondence: angela.fajsi@uns.ac.rs; Tel.: +381-60-446-47-13

**Abstract:** Even though Industry 4.0 is primarily focused on the implementation of advanced digital technologies, this is not the only aspect that should be considered. One of the aspects that calls for attention is the ability to create a sustainable and agile industrial environment. In this sense, the role of project management is crucial for achieving business excellence in a new industrial paradigm. The main goal of this paper was to determine the impact of different levels of project management maturity on business excellence in the context of Industry 4.0. The research in the paper was made using a sample of 124 organizations, differing in industry type and size, and recognized through the business excellence awards or recognitions given by European Foundation for Quality Management (EFQM). Using the Project Management Maturity Model (ProMMM), a significant connection was found between project management maturity and business excellence. Considering technology advances, these relationships were further examined in the context of Industry 4.0. Empirically based conclusions were drawn, which contribute to the literature on project management and business excellence in the context of Industry 4.0. Practitioners can implement them for more effective project management with the intention of bringing excellence into the organization's operations and results. Additionally, they can be useful to help organizations better cope with changing technology trends.

**Keywords:** project management maturity; business excellence; Industry 4.0; EFQM

## 1. Introduction

Strict rules and requirements regarding the knowledge economy and the modern industrial paradigm make organizations strive towards higher business excellence levels. The 'quality management' paradigm is moving towards 'managing quality', which is the basis of the business excellence concept that organizations strive for. Porter and Tanner [1] stated that 'the concept of business or organizational excellence provides support for the absolute integration of improvement initiatives within the organization'. It is based on the philosophy of continuous improvement, directing all organization's activities to enhance business performance, stakeholder satisfaction, corporate social responsibility, and environmental protection [2,3]. Toma and Marinescu [4] stated that there is a growing interest among companies in implementing business excellence strategies, which lead to increased quality of their business philosophy and improved business performances [4,5]. Effective formulation and implementation of these strategies have motivated organizations to change their way of doing business, and in this respect, to adopt various tools, methods, and techniques, such as enterprise resource planning (ERP) [4,6], balanced scorecard [7,8], lean or six sigma practices [9–11], and project management approaches [12–16], etc.

According to Kerzner [12], one of the main characteristics of organizations that were awarded the prestigious Malcolm Baldrige Business Excellence Award, was the existence of a project management system, which indicates a strong relationship between project management and business excellence. Effective project management at the organisation level does not just involve the application of software or the use of a specific tool [17].

To effectively implement this practice, which is thought to deliver sustainable project results, it is necessary to have acceptance and a positive attitude towards the project approach at all levels in the organisation, followed by the establishment of stable and long-term processes and competencies that will support its implementation and ensure excellence in their operations and results.

A key component of today's economy is a greater reliance on intelligence and intellectual abilities, rather than physical or natural resources [18], which contributes to the accelerated pace of scientific and technological progress related to Industry 4.0. This concept, also known as the fourth industrial revolution, helps "in implementing innovative technologies to improve productivity and working system" [19]. Jally et al. [19] also stated that the approaches to managing a project will be significantly altered due to the creation of these changes. The central aspect of the implementation of Industry 4.0 is the initiation of "smart business" and the acceleration of innovations through continual advancements where projects have a crucial role. Bag et al. [20] highlighted the role of project management in the process of Industry 4.0 integration and in achieving sustainable business. Considering everything aforementioned, the following research question arises: how does project management maturity affect organizational business excellence in the context of Industry 4.0?

The main goal of the study was to examine the impact of different levels of project management maturity on organizational business excellence in the context of Industry 4.0. For this purpose, an online questionnaire was distributed to organizations awarded or recognized for business excellence by the EFQM. The purpose of the questionnaire was to examine the levels of project management maturity in these organizations and their relationships with business excellence, considering their Industry 4.0 readiness. The ProMMM methodology [21] was used to assess the maturity level of project management, and the maturity level of Industry 4.0 was examined using the attributes defined by the authors Schumacher et al. [22].

In the next section, the theoretical background for project management and business excellence in the context of Industry 4.0 will be covered. Section 3 includes quantitative research, which encompasses a sample of 124 respondents and assesses the impact of project management maturity on business excellence and considers the readiness level for Industry 4.0 as a mediation variable. This section is concluded with a discussion to summarize and evaluate the research results. In the final section, managerial implications are presented, with potential limitations of the study and recommendations for future research.

## 2. Theoretical Background

### 2.1. Evaluating the Relationship between Project Management Maturity and Business Excellence

According to the EFQM [2], excellent organizations 'achieve and sustain outstanding levels of performance'. The study introduced by Talwar [23] acknowledges a positive relationship between business excellence implementation and organizational performance. Nowadays, several models are used to measure business excellence within an organization. The most cited models in the literature are the Malcolm Baldrige Criteria for Performance Excellence (CPE) and the European Foundation for Quality Management (EFQM). These models are based on TQM principles, and they cover topics such as customer focus, leadership, people involvement, continuous improvement, etc. The purpose of any business excellence model is to help organizations to sustain flexibility and embrace changes that could have a positive impact on their competency in the digital business environment. Achieving excellence in business activities implies adopting Deming's continuous improvement approach: plan, do, check, and act [24]. Many management practices support this approach. For example, project management [12–16] can be seen as a complementing part to the organisation's practice while reaching business excellence [1], even in the uncertain conditions that characterize technological changes. Planning, implementing, and controlling changes effectively are crucial in the process of implementing continuous improvements within organizations that aim to achieve business excellence. Vora [25] stated that only 30%

of organizational change programs are considered successful. One of the main reasons why most change management efforts fail is ineffective project management [25].

Project management is essential in today's business world—it is an approach that promotes continuous improvement through different types of projects that lead to improved organizational performance [26]. As already noted, Kerzner [13] emphasized the strong relationship between project management and business excellence, indicating that all organizations involved in the research which won the Malcolm Baldrige Award for Excellence also had a high level of project management implemented. In addition, Craddock [14] has proved that project success and sustainability are directly related to business excellence. As the business excellence models are based on the TQM principles, it is important to make links between TQM and project management approaches. The TQM is a fundamental concept of continuous improvement, within which organizations constantly review and enhance their business processes. Bryde and Robinson [27] emphasized that the TQM principles are important for maintaining effective project management, especially in customer service, failure prevention, professional development of employees, and strong leadership.

To measure an organization's project management effectiveness, different project management maturity models can be deployed. According to Kerzner [13], maturity in project management can be defined as 'the development of systems and processes that are repetitive in their nature to provide a high probability that each project will be successful'. The most used maturity models in the literature are the PMMM model [28], PM2-Project Management Process Maturity Model [29], and the Kerzner's project management maturity model [30]. On the other side, some models have moved from a strict relationship between CMMI and PMBOK group processes. Pennypacker and Grant [31] stated that one of these models is the ProMMM model [21], which is also based on the CMMI model, but instead of PMBOK elements, relationships from the EFQM Model are taken. Most existing models test the maturity of the project management processes, while using this model, organizations assess other attributes and provide a true picture of their project management capability. Therefore, the ProMMM model has a wide application in practice and empirical studies [32–35].

Achieving a satisfactory level of maturity is a continuous and long-term process. However, due to built-in constraints and environmental factors, many organizations are not able to reach the highest levels of maturity during their existence [21]. Andersen and Jessen [35] stated that fully matured organizations do not exist in the real world, so considering different levels of maturity is a reasonable task for any organization. Research presented by Backlund et al. [36] revealed that higher levels of project management maturity led to success in project implementation, which further leads to improved organisation's processes in their road to bring excellence [37]. Therefore, the main hypothesis is proposed:

**H1.** *A high level of project management maturity has a positive impact on business excellence.*

In modern business conditions, the area of project management faces a much more complex and dynamic environment as a characteristic of the new industrial revolution more generally known as Industry 4.0 [38].

### 2.2. Project Management Maturity and Business Excellence in the Context of Industry 4.0

Determining a relationship between project management maturity and business excellence is a complex issue affected by many factors related to Industry 4.0 and digitalization that comes with it. Raj et al. [39] opined that there is a growing need for implementation of standards and government regulations to accelerate the process of adoption of Industry 4.0 digital technologies. They also asserted that the "lack of a digital strategy alongside resource scarcity" followed by a "lack of standards, regulations, and forms of certification", constrains companies from strengthening their capabilities in the process of fully leveraging Industry 4.0 digital technologies. This concept is especially applicable to the manufacturing and IT industry, while Al Amri et al. [40] stated that its applicability to measure was still

uncertain for other areas. On the contrary, there are studies that confirm the importance of Industry 4.0 for service organizations [41,42].

The modern business excellence paradigm is strongly oriented 'to the necessity to transform the current organization for the future' [3]. Gunasekaran et al. [43] stated that it is important to define 'what might be the future of excellence'. The term 'future' relates in this context to digital transformation, Industry 4.0, and organizational agility with special emphasis on technology and human capacity development. Fonseca [44] made the comparison between the EFQM 2013 and EFQM 2020 models and stated that the new model has "a focus on the futuristic requirements of the organizations rather than merely a business excellence model and/or just a quality award enablement model".

According to the EFQM 2020 model, both concepts of business excellence and Industry 4.0 share a common goal to improve organizational operations and results. The Singapore Smart Industry Readiness EDB report [45] indicated that business excellence is directly related to human resource ability to adopt a range of different approaches, methods and tools promoted within Industry 4.0.

Industry 4.0 promotes the adoption of new organizational models but also the adaptation of existing ones to achieve excellence in the conditions set by the new industrial paradigm. Accordingly, project managers are looking for different ways to understand technological change and its impact on project management processes. Moreover, the role of project management in the development of Industry 4.0 is essential for its success and vice versa [46]. Therefore, the authors state that traditional project management systems should be analysed and updated according to the requirements of the new industrial revolution, which will help reduce the complexity of projects [21] and increase the likelihood of projects succeeding.

The above discussions lead to the following hypothesis:

**H2.** *Industry 4.0 readiness level is a mediator between project management maturity and business excellence.*

Due to a lack of research on the relationship between project management maturity and business excellence in the context of Industry 4.0, this issue requires further empirical analysis.

## 3. Research Methodology

### 3.1. Data Collection and Sample

Data collection began in January 2021 and continued through March 2021. The questionnaire was distributed in electronic format, via the Google Forms platform to organizations that have received awards and recognitions for business excellence [47]. Besides, an invitation letter and a survey were sent to the National Representatives for EFQM, so that they would be aware of the research to influence their members to participate in it.

The EFQM 2020 model was launched in November 2019 and when data collection began there was only a small number of organizations that followed the new model framework. For this reason, the sample included organizations that have achieved awards and recognitions for business excellence according to the EFQM 2013 model. It was emphasized that the survey should be filled in by a person who deals with project management or development processes. Their task was to assess the project management maturity and Industry 4.0 readiness levels within their organization.

The total number of participants who took part in the research was 130. Of those, six had missing data, so the final number of participating organizations was 124. The total number of relevant research organizations as of January 2021 was 1293, thus the response rate was 10.05%. Rogelberg and Stanton [48] stated that a response rate of 10% should not be ignored; rather it should be examined as to whether it has a substantial impact on the conclusions, considering that a lower response rate is important to understand topics that are insufficiently researched in the literature.

The greatest portion of respondents were employed in top management positions (37.90%), followed by middle-management (27.42%) and project management (11.29%). The participating organizations varied in size (Figure 1) and originated from 27 countries (Table 1). The following table shows the respondents by the type of activity they performed (Figure 2).

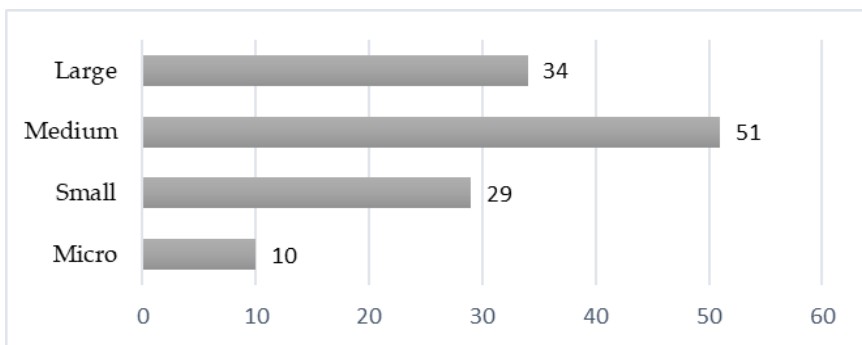

**Figure 1.** Profile of organizations by size.

**Table 1.** Profile of organizations by country.

| Country | N | % | Country | N | % |
|---|---|---|---|---|---|
| Spain | 28 | 22.58 | Finland | 2 | 1.61 |
| Switzerland | 13 | 10.48 | Greece | 2 | 1.61 |
| United Kingdom | 11 | 8.87 | Netherlands | 2 | 1.61 |
| Turkey | 9 | 7.26 | Jordan | 2 | 1.61 |
| Austria | 6 | 4.84 | Hungary | 2 | 1.61 |
| Portugal | 6 | 4.84 | Saudi Arabia | 2 | 1.61 |
| Ireland | 5 | 4.03 | Sweden | 2 | 1.61 |
| Germany | 5 | 4.03 | United Arab Emirates | 1 | 0.81 |
| Belgium | 4 | 3.23 | Italy | 1 | 0.81 |
| Ecuador | 4 | 3.23 | Peru | 1 | 0.81 |
| Colombia | 3 | 2.42 | Poland | 1 | 0.81 |
| Czech Republic | 3 | 2.42 | Russia | 1 | 0.81 |
| France | 3 | 2.42 | Slovenia | 1 | 0.81 |
| Lithuania | 3 | 2.42 | Missing data | 1 | 0.81 |

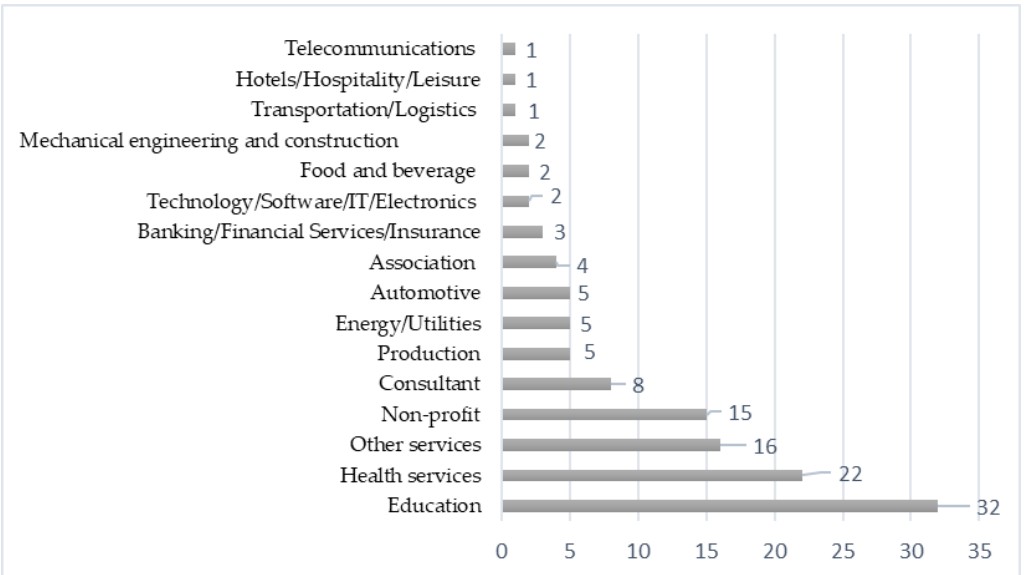

**Figure 2.** Profile of organizations by industry type.

Looking at the profile of participating organizations, most of them were medium-sized (41.13%) and had their headquarters in Europe because the EFQM model is most represented there. Most respondents were from Spain (22.58%), which had the most organizations with EFQM awards, followed by Switzerland (10.48%) and the United Kingdom (8.87%).

When it comes to the type of industry, most responses were from organizations engaged in education (25.81%), followed by health services (17.74%) and other service activities (12.90%). These numbers can be explained by the fact that these sectors/areas had the largest number of organizations with recognition for business excellence.

### 3.2. Research Instruments

### 3.2.1. Project Management Maturity

The aforementioned literature review showed different models for measuring project management maturity. The ProMMM model is based on the CMMI model [49] and the EFQM model was used to measure project management maturity within the organization. It describes four levels of project management maturity: Naïve, Novice, Normalized and Natural. They were further defined in terms of four attributes: culture, process, experience, and application. Each of those dimensions contained a set of five items that were measured using a 5-Point Likert scale. There was no significant difference between the average mark of the dimensions of project management maturity. Research has shown that organizations that are excellent in business have a slightly more developed project management culture compared to other attributes (average mark: culture 2.91, processes 2.79, experience 2.53, and applications 2.67).

### 3.2.2. Business Excellence

Business excellence levels were defined according to the prescribed criteria established by the EFQM, which were contained within the EFQM Excellence Model, defined as a 'framework for measuring the strengths and areas for improvement of an organization across all of its activities' [2]. The EFQM process recognition is a complex assessment. It is carried out by independent EFQM assessors, and organizations that have won EFQM recognition were taken as a sample. Different levels of recognition were presented in the form of the 7-Point Likert Scale. The majority of respondents were in the category Recognized for Excellence with 4 stars (24.19%), followed by Committed to Excellence (19.35%), Recognised for Excellence 5 stars (16.94), Committed to Excellence 2 stars (15.32%), Recognised for Excellence 3 stars (9.68%), EFQM Award Finalist (8.06%), EFQM Excellence Award/Prize Winner (6.45%).

### 3.2.3. Industry 4.0 Readiness

Industry 4.0 readiness was assessed based on two sets of questions:

(1) Stages of technological development were measured by a 4-Point Likert scale. A total of 32.26% of organizations stated that they used only existing, well-established, and mature technologies, and the same percentage of organizations stated that they used many new and recently developed technologies. Limited new technology, or 'a new feature', were used by 30.64% of respondents, while new, unproven technological concepts were used by only 4.84% of respondents.
(2) Dimensions and items of the Industry 4.0 Readiness model were measured by a 4-Point Likert scale (Table 2) [22].

### 3.3. Research Results and Discussion

### 3.3.1. Preliminary Analysis

The objectives of the preliminary analysis were to check the reliability of measures and to obtain insights into the dataset. Internal structure validity, reliability analysis and descriptive statistics were done for the purpose of this research.

To examine the internal structure of the test and the reliability of individual dimensions of project management maturity, analyses were performed on culture, processes, experience, application, exploratory factor analysis, and reliability.

The reliability of those dimensions was adequate ($\alpha > 0.70$) [50]. When it came to the Process dimension, it was noticeable that the item examining the degree of formality of the project management process had a very low loading. Removing this item would lead to an increase in dimension reliability. As expected, both an exploratory factor analysis and a reliability analysis showed increased values of relevant coefficients after excluding this dimension. The results of the exploratory factor analysis and the reliability analysis for the Industry 4.0 Readiness dimension were assessed as adequate.

**Table 2.** Industry 4.0 Readiness model.

| Areas of Industry 4.0 | Does Not Exist or It Is at a Very Low Level (%) | Low-Level (%) | Medium-Level (%) | High-Level (%) |
|---|---|---|---|---|
| Industry 4.0 strategy | 31.45 | 23.39 | 33.06 | 12.1 |
| Leadership | 16.13 | 26.61 | 37.1 | 20.16 |
| Customers | 8.87 | 31.45 | 40.32 | 19.35 |
| Products and services | 8.06 | 30.64 | 41.13 | 20.16 |
| Operations | 8.87 | 37.1 | 45.16 | 8.87 |
| Culture | 8.06 | 34.68 | 38.71 | 18.55 |
| People | 8.06 | 31.45 | 45.97 | 14.52 |
| Governance | 12.90 | 31.45 | 44.35 | 11.3 |
| Technology | 8.06 | 33.87 | 41.13 | 16.93 |

### 3.3.2. Descriptive Statistics

The parameters distribution shape, skewness, and flatness, showed that the distribution had the typical bell curve pattern of normal distributions (Table 3). The normal distribution, according to the conventional criterion, has the value of the stated parameters in the range $\pm 1.5$ [51].

**Table 3.** Descriptive statistics.

| Variable | Min | Max | AS | SD | Sk | Ku |
|---|---|---|---|---|---|---|
| Culture | 1 | 20 | 14.63 | 3.70 | −0.89 | 1.27 |
| Processes | 4 | 20 | 13.98 | 3.17 | −0.46 | 0.29 |
| Experience | 0 | 20 | 12.71 | 3.91 | −0.61 | 0.50 |
| Application | 0 | 20 | 13.42 | 3.78 | −0.83 | 0.67 |
| Business excellence | 1 | 7 | 3.56 | 1.82 | 0.08 | −1.00 |
| Industry 4.0 readiness | 10 | 38 | 25.54 | 6.28 | −0.27 | −0.42 |

Legenda. Min—minimum value. Max—maximum value. AS—arithmetic mean. SD—standard deviation. Sk—skewness. Ku—kurtosis.

### 3.3.3. Mediation Analysis—Effects of Project Management Maturity on Business Excellence in the Context of Industry 4.0

Hayes' macro 'process' v4.0 software [52] was used to test the mediation effect of Industry 4.0 readiness on the relationship between project management maturity and business excellence. A conceptual diagram of mediation analysis was shown in Figure 3.

The analysis was conducted using 5000 bootstrap samples and with 95 confidence intervals, in line with Hayes's [52] suggestion. Overall, the mediation model was significant ($F (2, 121) = 5.36$, $p < 0.01$, $R^2 = 0.081$). Individual relations between variables are presented in the Table 4.

Significant effects were found for the a-path (direct effect from Project management maturity on Industry 4.0 readiness) and the c'-path (direct effect from Project management maturity on Business excellence). On the other hand, the c-path (indirect effect of Project management maturity on Business excellence through Industry 4.0 readiness) and the

b-path (direct effect from Industry 4.0 readiness to Business excellence) were not significant. As the results suggested:

**H1.** *A high level of project management maturity has a positive impact on business excellence—was accepted.*

**H2.** *Industry 4.0 readiness level is a mediator between the project management maturity and business excellence—was rejected.*

In addition to providing better explanations for these relationships, the authors examined which Industry 4.0 technologies have been used within respondent organizations. The discussion chapter will include the importance and relevance of these findings.

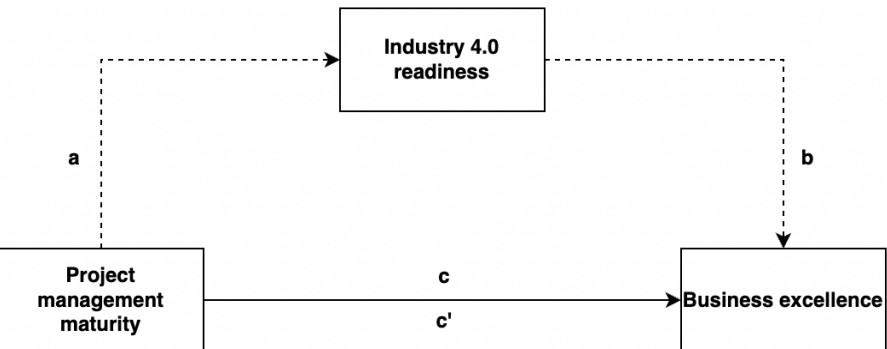

**Figure 3.** Conceptual diagram of mediation analysis.

**Table 4.** Description of individual relations between variables.

| Path | Description | Parameters |
|------|-------------|------------|
| a | Direct effect from Project management maturity on Industry 4.0 readiness | $\beta = 0.28$; 95 CI = 0.106–0.452 |
| b | Direct effect from Industry 4.0 readiness to Business excellence | $\beta = 0.04$; 95 CI = −0.003–0.111 |
| c | Indirect effect of Project management maturity on Business excellence through Industry 4.0 readiness | $\beta = 0.04$; 95 CI = −0.003–0.111 |
| c′ | Direct effect from Project management maturity on Business excellence | $\beta = 0.19$; 95 CI = 0.056–0.373 |

## 4. Discussion

This study presents empirical evidence linking project management maturity and business excellence in the context of Industry 4.0. The hypothesis, which claims that a higher level of project management maturity in organizations has a positive effect on business excellence, was confirmed by using Hayes' macro "process" v4.0 software. Business excellence is achieved through continuous improvement, innovation, and learning [2], and, importantly, the project management approach is in line with those principles [53–55].

There were no statistically significant differences between the individual dimensions of project management maturity and their impact on business excellence. All dimensions had almost the same effect in synergy, and therefore, organizations should understand and develop project management culture, establish processes, educate people, and effectively apply project management methods and tools. Nevertheless, culture has a slightly higher level of maturity compared to other attributes (2.91), which indicates that excellent organizations clearly define and support the "corporate culture" for project management [56].

In this study, there was no evidence of a mediating effect of Industry 4.0 readiness on the relationship between project management maturity and business excellence. Strong statistical significance was found for the effect of project management maturity on Industry 4.0 readiness, which is in line with previous studies [19,57–61]. The Industry 4.0 variable

had no statistically significant impact on business excellence, which contradicts previous studies [43,44,46]. The authors explain that this was due to the specificity of the sample, because mainly the organizations that were engaged in the service industry participated in the research. Although the literature proves the importance and necessity of applying the concept of Industry 4.0, industry-related institutes define its application, and the progress is very slow [34], especially when it comes to the service industry. Most previous studies covered the topic of Industry 4.0 in manufacturing companies [39,62], while Bodrow [41] and Rennung et al. [42] showed that the service industry becomes an important element of the Industry 4.0 concept, mainly for the reason that most products and services are connected into the integrative offering.

In addition, the advanced technologies used in organizations were examined. The use of more advanced programs in the IT system such as ERP, CRM, and the use of mobile technologies are mostly represented; they were used in almost 70% of organizations. Jally et al. [19] defined Industry 4.0 technologies that were significant for project management, such as additive manufacturing, IoT, Autonomous systems, Big data, which are presented in more than 20% of respondent organizations. These data indicate that more than 20% organizations have technologies which can be successfully integrated with the project management approach. Jun et al. [63] identified these technologies as important for useful and accurate quality management. Other technologies, such as artificial intelligence and blockchain, are less represented because they are primarily related to the manufacturing industry. The development of smart services is expected in the near future, and it is evident that these will not only influence but also facilitate project management within the organization, with the aim of achieving and/or maintaining business excellence.

### 4.1. Theoretical Implications

This study aimed to test the theory of project management that links the level of project management maturity with business excellence in the context of Industry 4.0. It has been proven that project management has a strong positive impact on business excellence, and it provides empirical evidence in theory that previous studies support. Furthermore, the model developed for this research raises the possibility for other researchers in the field to incorporate specificities into their studies of the new industry trends imposed by Industry 4.0.

### 4.2. Practical Implications

In a practical manner, the findings can help organizations to define strategies for more effective implementation of project management approach to achieve and/or maintain business excellence within the new industrial paradigm.

The finding suggests that a balanced development of project management culture, processes, people, methods, and tools for application leads to excellence in business operations and results. Adopting an organization's project management culture helps organizations to understand and adapt their core activities to different norms, regulations, and behaviours. Furthermore, it influences employee's expertise and commitment, project management processes and its application by using a variety of methods and tools such as requirement analysis, timeline frameworks, agile methods, specific software to support project management, etc. [64].

The literature review found that technologies, such as additive manufacturing, IoT, Autonomous systems, and Big data, have a positive impact on both project management and business quality management, which indicates to practitioners the importance of their more intensive use.

As a final practical implication, the authors suggest that organisations operating in emerging economies, where the conditions for achieving awards for excellence have not yet been met, should consider applying the modified ProMMM model. With several limitations, primarily caused by unfavourable environmental factors, the proposed model can provide

a clear direction regarding the organization of project activities to improve their business results, stakeholder satisfaction, socially responsible business, and environmental protection.

## 5. Conclusions, Limitations and Future Research

The effective deployment of project management approach helps organizations to deal with the issues of achieving business excellence. A novel modified ProMMM model was proposed to access project management maturity within organizations. The empirical evidence found in this study shows that higher levels of project management maturity led to more recognition and awards for business excellence.

Examining the mediation role of Industry 4.0, no significant statistical differences were observed. This contradicts previous studies, which have emphasized the importance of Industry 4.0 in the context of project management and business excellence, but there are no studies that have examined Industry 4.0 as a mediation effect in the aforementioned relationship.

One of the major limitations of this study was the limited number of participants in the research. Given that the sample consisted of a specific population of respondents that included organizations that exclusively have some form of EFQM recognition, it is considered that a sample of 100 or more respondents is acceptable for valid results [65,66]. Additionally, the invitation letter invited respondents engaged in project management or development processes, which further narrowed the sampled population. It is necessary to involve a larger number of people from organizations where the maturity of the process will be viewed from different aspects. The creator of the original questionnaire suggested that information needs to be obtained from a wide range of staff to avoid responses from specific individuals [21]. This limitation can be overcome by implementing qualitative methods that can perform a more detailed analysis and verify the results obtained in quantitative research. Hillson [21] suggested methods such as interviews and case studies.

The authors emphasized that there is a need for further development in this area through empirical studies on this topic, especially including manufacturing companies that are closely related to Industry 4.0 practices. Further researches can investigate relationships between project management maturity and business excellence model dimensions (such as leadership, strategy, customers, etc.) to determine the level of project management impact on individual dimensions. Furthermore, Malcolm Baldrige Criteria for Performance Excellence (CPE) can be used as criteria for some future research to verify results obtained in this research where the EFQM model was used.

**Author Contributions:** Conceptualization, A.F., S.M. and N.M.; methodology, A.F., S.M. and N.M.; validation, S.M.; formal analysis, A.F.; investigation, A.F. and S.M.; resources, S.M.; data curation, M.M.; writing—original draft preparation, A.F.; writing—review and editing, M.M., S.M. and N.M.; visualization, A.F. and M.M.; supervision, S.M.; project administration, A.F. All authors have read and agreed to the published version of the manuscript.

**Funding:** This research received no external funding.

**Acknowledgments:** The results presented in this paper are part of the research within the project "Improvement of teaching processes at DIEM through the implementation of the results of scientific research in the field of Industrial engineering and management", Department of Industrial Engineering and Management, Faculty of Technical Sciences in Novi Sad, University of Novi Sad, Republic of Serbia.

**Conflicts of Interest:** The authors declare no conflict of interest.

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
