# Peer review of "Project Management Maturity and Business Excellence in the Context of Industry 4.0"

_processes, doi:10.3390/pr10061155_

Round 1

Reviewer 1 Report

I see selected topic as very important issue. Defining project management role in Industry 4.0 transformation is very important issue. Only about 11% of transformations are completed with success. That gives us clear view of some critical issues connected with transformation success.

Please check information about OPM3. If I remember ocrrectly, that model is no longer supported by PMI.

My major concern is connected with H1. For me it is quite obvious that highe project management maturity (so better project delivery, better value from projects, better strategic alignment etc.) will lead us to business excellence.

Author Response

Dear Reviewer,

We hope this letter finds you well.

Thank you for the time and dedication you invested in the careful revision of this paper We have tried our best to revise our manuscript according to the comments we received from reviewers. Attached please find the point by point response to your comments.

Yours Sincerely,

Angela Fajsi on behalf of the authors

Reviewer 2 Report

The abstract is weak. Authors fail to define the objective of the study and what is a novelty in their study.

The introduction section is poorly organized; there are lots of missing links, and in addition, the problem should be explained based on the necessity of researching the current subject in the introductory section, which does not happen in the current format. This section should be completely revised.

The literature review is also poorly organised. I have never seen any research gap as a subsection, which is very important for proving the novelty of the paper. 

Many papers have been published in this area such as https://doi.org/10.1016/j.ijpe.2019.107546 that needs to update in the introduction and literature review section. 

The author needs to compare their proposed model with others in order to elevate their importance and urge towards this study.

The conclusion section looks like a summary, there are no in-depth insights in this section, and the managerial implications are also not addressed.

Author Response

Dear Reviewer,

We hope this letter finds you well.

Thank you for the time and dedication you invested in the careful revision of this paper, your constructive comments and suggestions are much appreciated. It is our belief that the manuscript is substantially improved after making the suggested edits.

We have tried our best to revise our manuscript according to the comments. Please find attached point by point response to your comments.

Yours Sincerely,

Angela Fajsi on behalf of the authors

Round 2

Reviewer 2 Report

none